# Integrated Analysis of Metabolomics, Flavoromics, and Transcriptomics for Evaluating New Varieties of *Amomum villosum* Lour.

**DOI:** 10.3390/plants13172382

**Published:** 2024-08-26

**Authors:** Zhenkai Li, Xin Luo, Yanli Yao, Yukun Wang, Zhiheng Dai, Tianle Cheng, Xinzhi Huang, Mei Bai, Junjun He, Hong Wu

**Affiliations:** 1State Key Laboratory for Conservation and Utilization of Subtropical Agro-Bioresources, Guangdong Laboratory for Lingnan Modern Agriculture, College of Life Sciences, South China Agricultural University, Guangzhou 510642, China; lzk0530@163.com (Z.L.); baimei924@scau.edu.cn (M.B.); 2South Subtropical Crop Research Institute, China Academy of Tropical Agricultural Sciences, Key Laboratory of Tropical Fruit Biology, Ministry of Agriculture & Rural Affairs, Zhanjiang 524013, China; yaoyanli1983@163.com; 3Guangdong Provincial Key Laboratory of Utilization and Conservation of Food and Medicinal Re-Sources in Northern Region, College of Biology and Agriculture, Shaoguan University, Shaoguan 512005, China; 4Guangzhou Dublin International College of Life Sciences and Technology, South China Agricultural University, Guangzhou 510642, China

**Keywords:** *Amomum villosum* Lour., germplasm evaluation, metabolomics, flavoromics, transcriptomics

## Abstract

*Amomum villosum* Lour. (*A. villosum*) is the original plant of the medicinal and culinary spice Amomi Fructus (Sharen) and is an important economic crop in the Lingnan region of China. During the cultivation and production of *A. villosum*, prolonged reliance on single asexual reproduction has exacerbated the degradation of its varieties, leading to inconsistent yields and quality. Building upon earlier cultivar selection efforts, this study provides a comprehensive evaluation of two newly bred *A. villosum* varieties (A11 and A12) from perspectives including plant traits, product characteristics, active ingredients, and multi-omics analysis. It was found that A12 plants display enhanced robustness, more aromatic fruits, higher yields, and elevated levels of bornyl acetate, A11 shows the advantage of a high camphor content, and the different metabolites and differentially expressed genes of the two varieties were significantly enriched in multiple metabolic pathways. Additionally, A12 contained more terpenoids and substances with aromatic odors such as sweet, fruity, floral, and green. Furthermore, a key gene (Wv_032842) regulating the acetylation of bornyl was discovered, and its significantly higher expression, in A12. In conclusion, this study has a guiding significance for the evaluation of germplasm resources and the breeding of excellent varieties of *A. villosum*.

## 1. Introduction

In the natural world, certain special plants serve not only as sources of spices or edible foods but also as raw materials for medicinal purposes. Examples include *Cinnamomum cassia* (L.) D. Don, *Zanthoxylum bungeanum* Maxim., *Pogostemon cablin* (Blanco) Benth., *Lycium barbarum* L., and *Amomum villosum Lour. (A. villosum)* [1]. Their extensive applications endow them with increased market demand and utility value [2]. These original plants have transitioned from wild to cultivated production, evolving into distinctive economic crops. Breeding different varieties is crucial for enhancing crop yield and quality, driving agricultural innovation, and sustainable development [3]. At the species level of these original plants, through artificial selection and cultivation, diverse varieties with varying characteristics can be obtained, enriching crop diversity and meeting diverse market demands [4]. However, compared to vegetables and fruits, the breeding of dual-purpose medicinal and edible crops is more challenging and progresses more slowly, often requiring higher standards of effectiveness and safety [5].

Amomi Fructus (known as “Sharen” in Chinese) is a culinary spice renowned for its distinctive flavor and is also esteemed as a valuable traditional Chinese medicinal herb [6,7]. *A. villosum* serves as the botanical origin of Amomi Fructus [8], with its dried mature fruits and seed clusters constituting the primary edible and medicinal parts [8]. It is utilized not only directly for seasoning and consumption but also finds favor as a key ingredient in teas, preserved fruits, beverages, and other products [9]. Research indicates that Amomi Fructus possesses beneficial effects including gastroprotective, anti-inflammatory, antimicrobial, and hepatoprotective properties, attributed to its content of bornyl acetate, camphor, borneol, and various biologically active compounds such as flavonoids, alkaloids, and polysaccharides [10,11]. Due to its extensive applications and significant market value, *A. villosum* has emerged as a crucial economic crop in southern China [12].

However, the sustainable development of the *A. villosum* industry has long been hindered by issues such as low yields and uneven quality levels, closely intertwined with the genetic resources, growth environments, and cultivation techniques of *A. villosum*. *A. villosum* is a tropical or subtropical rainforest plant with strict and specific requirements for its growth environment. It is distributed in the tropical or subtropical monsoon climate areas ranging from 99.56° E to 112.26° E and 21.27° N to 23.27° N. It prefers to grow in fertile, moist, shaded valleys, with a relative humidity requirement of over 90%, avoiding waterlogging and cold, and thriving in temperatures between 19 °C and 22 °C; temperatures below −3 °C will result in death [13,14]. Yangchun City in Guangdong Province is the original birthplace of *A. villosum* and remains the most suitable high-quality production region for its cultivation [15]. *A. villosum* produced here exhibits superior quality but is characterized by low yield, averaging only 150 kg·hm^−2^ of dried fruit. Despite extensive cultivation efforts implemented since the 1950s in Yunnan, Guangxi, and other regions with the aim of boosting production, the relocation of cultivation sites has led to significant changes in the clinical efficacy and concentration of active ingredients in Amomi Fructus [15,16,17]. As a result, noticeable distinctions persist between the *A. villosum* produced in Yangchun City and that cultivated elsewhere. Presently, the demand for *A. villosum* from Yangchun City continues to outstrip supply, leading to high prices, exceeding 6000 RMB per kilogram, which is 5 to 10 times higher than the prices in other production areas. Consequently, addressing the challenge of achieving high yield and quality production within limited suitable habitats, while simultaneously developing new germplasm resources and production technologies to enhance its adaptability to diverse environments, is crucial for the sustainable development of the *A. villosum* industry.

Germplasm resources are the crucial factors determining the quality and yield of crops, and excellent varieties form the essential foundation for achieving high quality and productivity [18,19]. However, variety breeding has always been a weak link in the basic research and industrial development of *A. villosum*. In particular, as the primary breeding method for *A. villosum*, asexual reproduction has led to serious problems such as seed source contamination and strain degradation due to long-term blind introduction and a single genetic background [20]. This has been a significant factor contributing to the instability of *A. villosum* quality, low yields, and susceptibility to pests and diseases. Therefore, strengthening the protection of *A. villosum* germplasm resources, developing new varieties, and selecting high-quality and high-yielding *A. villosum* varieties is crucial for the sustainable development of the *A. villosum* industry [21].

Based on the aforementioned research background, our research group has conducted long-term purification and breeding of wild-type *A. villosum*, resulting in the development of two new varieties, “Zhan Sha 11” (A11) and “Zhan Sha 12” (A12), which exhibit good stability, consistency, and distinct traits. Both varieties obtained certification as new varieties in May 2023 (A11 certificate number: CNA20201000982; A12: CNA20201004913). Specifically, A11 represents a widely characteristic variety within the *A. villosum* population, whereas A12 displays specific traits observed in a minority of the population. To comprehensively evaluate these two varieties, this study employed DNA barcoding analysis for molecular identification and compared their characteristics from various aspects including plant morphology, product traits, and active ingredients. Furthermore, metabolomics, flavoromics, and transcriptomics technologies were utilized to explore the underlying mechanisms contributing to their differences, aiming to provide guidance for the evaluation of *A. villosum* germplasm resources and the breeding of superior varieties.

## 2. Results

### 2.1. DNA Barcode Analysis

In order to ensure the authenticity of the germplasm, referencing the “DNA Barcode Standard Sequence for Chinese Pharmacopoeia Herbal Medicines”, the standard DNA barcode for *A. villosum* was published and used to identify the two *A. villosum* varieties. Additionally, their genetic distance from closely related species and counterfeit original plants was analyzed. The results showed that the obtained Internal Transcribed Spacer 2 (ITS2) characteristic sequences for both varieties were highly similar to the *A. villosum* standard sequence (Appendix A). BLAST alignment in the NCBI database also demonstrated a high degree of matching with *A. villosum*. Genetic distance analysis based on the ITS2 sequences revealed that the genetic distance between the ITS2 sequences of *A. villosum* sources was smaller, all less than 0.01, while they exhibited significant genetic distance from five other plant species (Figure 1A). The phylogenetic tree also indicated that the original plants of the counterfeit products formed a distinct group, while the closely related plants and *A. villosum* ITS2 sequences formed separate clusters (Figure 1B). This suggests that both varieties can serve as the original plants of Fructus Amomi and are closely related.

### 2.2. Comparison of Plant Traits and Agronomic Characteristics

To analyze the characteristics of the two *A. villosum* varieties, a comprehensive comparison of the medicinal properties and plant morphology of A11 and A12 was conducted. The study results indicate that both A11 and A12 exhibit the fundamental traits of *A. villosum* plants and medicinal materials. However, they also display certain differences. These differences are primarily observed in the seed clusters, seeds, fruits, flowers, root systems, leaf count, and plant height (Figure 2, Appendix A). Specifically, the A12 plants are taller on average (plant height: 254 cm), possess more leaves (leaf count: 36), are sturdier (stem diameter: 12.79 mm; rhizome diameter: 6.89 mm), have four to five rhizome branches at the base, and produce more inflorescences (20 Pcs·m^−2^) that are larger (spike length: 19.7 cm; peduncle length: 13.3 cm). The seed clusters and fruits of A12 are nearly spherical, with denser spines, fewer seeds per fruit (seed count: 25), and a higher thousand-seed weight (19.2 g). In contrast, A11 has an average plant height of 202 cm, an average leaf count of 29, typically three rhizome branches at the base, and nearly oval-shaped seed clusters and fruits with relatively sparse spines. The average seed count per the A11 fruit is 46, with a thousand-seed weight of approximately 12 g. In terms of yield, A12 exhibited a higher dry fruit yield at 479.58 kg·hm^−2^, whereas A11 showed a lower dry fruit yield at 447.61 kg·hm^−2^. Additionally, A12 possesses a more aromatic scent and sweeter taste, whereas A11 exhibits a more sour and astringent flavor.

### 2.3. Differences in the Content of Major Active Ingredients

The volatile oil content and the content of three main active components (bornyl acetate, camphor, and borneol) in the seed clusters of both *A. villosum* varieties were determined. The results showed that there were no significant differences in the total volatile oil content and borneol content between the two varieties. However, the content of bornyl acetate in A12 (12.50 mg·g^−1^) was significantly higher than that in A11 (9.89 mg·g^−1^), while the content of camphor was the opposite, with A11 (5.34 mg·g^−1^) significantly higher than A12 (2.62 mg·g^−1^) (Figure 3). This reflects a difference in the content of the major active components between the two varieties.

### 2.4. Metabolomics Analysis Results

By utilizing two platforms, LC-MS/MS and GC-MS, a comprehensive metabolomic analysis was conducted on the metabolites of two *A. villosum* varieties. The obtained data were subjected to coefficient of variation (CV) analysis, revealing that substances with CV values less than 0.5 in both quality control (QC) samples and sample detections accounted for over 85% (Figure 4A), indicating the stability of the detection data. The total ion chromatography (TIC) profiles (Figure 4B,C) showed that the two *A. villosum* varieties exhibited similar GC and LC chromatographic profiles and characteristic peaks, but differences were observed in the size of certain peaks, suggesting potential variations in substance content between the two varieties. Upon importing the obtained mass spectrometry data into a self-built metabolite database, a total of 2570 metabolites was identified and categorized into 26 major classes, with terpenoids (378 kinds), flavonoids (269 kinds), and phenolic acids (267 kinds) representing the highest proportions (Appendix A, Figure 4D). Relative odor activity value (rOAV) analysis was performed on the detected compounds, using a rOAV ≥ 1 to assess their direct contribution to the flavor of the samples. The results (Figure 4F, Appendix A) revealed that 274 substances in the two varieties had a rOAV ≥ 1, predominantly consisting of terpenoids (67 kinds) and esters (53 kinds), including key active ingredients such as bornyl acetate, camphor, borneol, and β-caryophyllene. This indicates the significant roles these substances play in shaping the flavor of *A. villosum* and influencing the medicinal efficacy and quality of the herbal material. Additionally, principal component analysis (PCA) of the full spectrum of metabolites from the two *A. villosum* varieties demonstrated minimal variation within groups and considerable variation between groups, highlighting distinct metabolic profiles between the two varieties (Figure 4E).

To screen for differential metabolites between the two varieties, orthogonal partial least squares discriminant analysis (OPLS-DA) was performed on the full spectrum of metabolites to build an OPLS-DA model. The results showed that the two varieties clustered significantly into two classes (Appendix A), and 378 metabolites with Variable Importance in the Projection (*VIP*) > 1 were selected. Concurrently, differential fold change (FC) analysis was conducted on the metabolites of the two varieties, screening for metabolites with fold change ≥ 2 and fold change ≤ 0.5, and then combined with the *VIP* > 1 selection criteria to identify significantly different metabolites. A total of 381 significantly different metabolites were ultimately screened, including 211 significantly upregulated and 170 significantly downregulated in A12 compared to A11 (Figure 5A). These significantly different metabolites are displayed in a scatter plot, as shown in Figure 5B, with terpenoids being the most abundant among them. Among these terpenoid compounds, 23 substances including bicyclosesquiphellandrene (WMW0013*151) and longifolene (WMW0008) were significantly highly expressed in A12 (*p* < 0.05), while 12 substances including Foliasalacioside B1 (Lagp004602) and Kravanhin B (zjsp123101) were significantly highly expressed in A11 (Figure 5C).

### 2.5. Differential Metabolite Flavoromics Analysis Results

Based on the results of metabolomics analysis, further flavoromics analysis was conducted (Appendix A). The differential metabolites identified between A12 and A11 were annotated with sensory flavors. From these annotated flavors, the top 10 sensory flavors with the highest number of associated metabolites were selected for radar chart plotting (Appendix A). The results showed that substances related to sweet (22 kinds) were the most abundant, followed by fruity (17 kinds), floral (13 kinds), and green (13 kinds), which could be the main reasons for the taste differences between the two varieties of *A. villosum*. Flavoromics Sankey diagram analysis (Figure 6) showed that compared to A11, there were more metabolites with high expression in A12 annotated to sweet, fruity, floral, green, and other sensory flavors. These highly expressed flavor substances determined that A12 had a more intense aromatic odor. For example, 3-Phenyl-1-propanol acetate (NMW0161) can enhance the flavors of sweet, spicy, and balsamic, beta-Guaiene (KMW0582*151) can enhance the flavors of sweet, woody, and spicy, and 3-Buten-2-one, 4-(2,2,6-trimethyl-7-oxabicyclo[4.1.0]hept-1-yl) (KM0639) can decrease the flavors of sweet, fruity, and woody.

### 2.6. Transcriptomic Analysis Results

In parallel with the metabolomic analysis, transcriptomic analysis was conducted on all the samples using the publicly available *A. villosum* genome as a reference genome [22] and utilizing FPKM (fragments per kilobase of transcript per million fragments mapped) as the metric for assessing gene expression levels (detailed sequencing results are provided in Appendix A). By aligning with the reference genome, a total of 23,617 genes were identified and annotated. As shown in the violin plot of gene expression levels in Figure 7A, the overall gene expression levels of all the samples were at a similar level, with no significant outlier samples, meeting the requirements for differential gene expression analysis. Principal component analysis (PCA) (Figure 7B) showed that samples of the same variety were more clustered, while samples of different varieties were relatively dispersed, indicating differences in gene expression between the two *A. villosum* varieties. Using DESeq2 for inter-group differential expression analysis of the two varieties, multiple hypothesis testing correction was performed using the hypothesis test probability (*p-value*) to obtain the false discovery rate (FDR). Differential genes were filtered based on the criteria of |log2 Fold Change| ≥ 1 and FDR < 0.05. The volcano plot in Figure 7C displays the filtering results, revealing that compared to A11, A12 had 950 upregulated genes and 1335 downregulated genes. Hierarchical clustering heatmap analysis of all the differential genes (Figure 7D) showed that samples of the two varieties were clustered separately, indicating that these differentially expressed genes reflect differences in gene expression between the two samples. Furthermore, KEGG pathway enrichment analysis was performed on the differentially expressed genes. Figure 7E shows the five KEGG pathways, with the smallest q-value in KEGG level 1 (if the enriched pathways in each KEGG level 1 are less than 5, all are displayed). Among them, Ko01110 (biosynthesis of secondary metabolites) had the most enriched genes, while Ko04141 (protein processing in the endoplasmic reticulum) showed the most significant enrichment. A12 had more upregulated genes in pathways such as Ko02010 (ABC transporters), Ko04814 (motor proteins), and Ko04626 (plant–pathogen interaction), while A11 had more upregulated genes in pathways such as Ko04141, Ko01110, and Ko04144.

### 2.7. Metabolome and Transcriptome Correlation Analysis Results

The differentially expressed genes and metabolites were subjected to KEGG pathway enrichment analysis. A total of 54 metabolic pathways were enriched by both sets, primarily within pathways such as biosynthesis of secondary metabolites, biosynthesis of various plant secondary metabolites, and flavone and flavonol biosynthesis. Particularly notable was the enrichment of 17 differentially expressed metabolites and 223 differentially expressed genes in the biosynthesis of secondary metabolites pathway (Figure 8A shows a bubble plot of the top 25 pathways ranked by differential gene *p*-values in the KEGG enrichment analysis). It is worth noting that among these 25 pathways, 21 were predominantly enriched in A12 (Figure 8B). The results of the KEGG pathway enrichment analysis further illustrate the differences between the two varieties in the synthesis and accumulation of secondary metabolites.

To further investigate the relationship between differentially expressed genes and metabolites, canonical correlation analysis (CCA) was performed on the enriched primary metabolic pathway (biosynthesis of secondary metabolites, ko01110). This analysis aimed to evaluate the correlations between metabolites and genes. The results are illustrated using a CCA plot (Figure 8C), which is divided into four quadrants. Within each quadrant, points that are further from the origin are more closely clustered together, indicating a higher degree of correlation and suggesting a more significant role within the overall metabolic pathway. Additionally, a correlation network analysis was conducted on the differentially expressed genes and metabolites within the biosynthesis of various plant secondary metabolites pathway (ko00999). The results, displayed in Figure 8D, reveal significant correlations between the metabolites S-(5′-Adenosyl)-L-methionine (pme2735), N-Feruloylagmatine (Zbzp002804), Hydrocoumarin (NMW0170), Scopoletin-7-O-glucoside (MWSmce024), and multiple genes (Wv_040025, Wv_020079, et al.) within the pathway.

### 2.8. Analysis of the Correlation between the Main Active Ingredients and Key Genes, and qPCR Validation

Borneol acetate, camphor, and borneol are important bioactive compounds in *A. villosum* [5,6]. Within *A. villosum*, borneol serves as a common substrate for the production of both borneol acetate and camphor. In this study, there was no significant difference in the borneol content between A12 and A11; however, the levels of borneol acetate and camphor showed significant differences. Consequently, we focused on the expression of key genes involved in the conversion of borneol to borneol acetate and camphor in the two varieties. By comparing the *A. villosum* genome [22], we screened the transcriptome analysis results of the two varieties and identified eight genes regulating borneol acetate synthesis and two genes regulating camphor synthesis. However, only five genes showed expression (Wv_032842, Wv_001248, Wv_006346, Wv_054143, and Wv_049056), all of which are involved in the regulation of borneol acetate synthesis. Among them, Wv_032842 was significantly upregulated in the A12 vs. A11 comparison, while the other four genes did not reach a significant difference level. The correlation analysis between the five genes and acetic acid borneol ester, camphor, and borneol revealed that Wv_054143, Wv_032842, Wv_001248, and Wv_006346 showed varying degrees of positive correlation with acetic acid borneol ester, while Wv_049056 exhibited a significant negative correlation (Figure 9A). The correlation analysis with camphor exhibited an opposite pattern compared to the acetic acid borneol ester. Further qPCR expression analysis of the five genes in the two varieties revealed that Wv_032842 exhibited significantly higher expression, while the other four genes did not show a statistically significant difference in expression levels (Figure 9B). Therefore, the differential expression of Wv_032842 may be the main reason for the significant difference in borneol acetate content between the two varieties.

## 3. Discussion

The development of different varieties often alters the quality, yield, flavor, efficacy, and physiological characteristics of dual-purpose medicinal and edible crops, thereby determining their distinct application prospects [23]. *Lycium barbarum* L. (Ningxia goji), a well-known medicinal herb and health food source of goji berries, has over 20 developed varieties. These varieties exhibit significant differences in physiological traits, commercial characteristics, major active ingredients (such as betaine and Lycium polysaccharides), and active effects [24,25]. Based on their characteristics, these varieties are applied in various fields including medicinal herbs, seasonings, tea, fresh fruits, and snacks. Different varieties of *Chrysanthemum morifolium* (Ramat.) Hemsl. display distinct chemical composition characteristics [26] and notable flavor differences; for instance, Huangju and Boju exhibit stronger astringency compared to Hangbaiju, which has a milder flavor. Yu et al. [27] found significant variations in the content of total phenols, rutin, and kaempferol glycosides among different varieties of *Zanthoxylum bungeanum*, with Fengxian Dahongpao exhibiting the strongest antioxidant activity. Various varieties of *Pogostemon Cablin* exhibit morphological diversity, as well as differences in chemical composition, active effects, and flavonoid metabolism regulation pathways [28].

This study conducted a germplasm evaluation of two varieties of *A. villosum* from multiple perspectives. As a member of the ginger family, *A. villosum* reproduces asexually through its rhizomes, which, upon maturation, give rise to erect plants, with new rhizomes sprouting from the base of the new plant, thus facilitating reproduction. Upon reaching a certain population size, the rhizomes produce inflorescences, and after pollination by insects, the small flowers develop into fruits. In this study, we found that compared to A11, the plants of A12 are taller, with stronger root systems, more extensive branching at the base of the rhizomes, with larger and more numerous inflorescences, making it easier for them to establish large populations. Additionally, they possess more sites capable of producing inflorescences, resulting in a higher yield, larger and more numerous inflorescences, and an increased ability to attract pollinating insects, thereby exhibiting superior traits.

Regarding fruit characteristics, A12 is similar in size to A11, but it has heavier seeds, a more intense aroma, and better taste, indicating its superior potential for food development and utilization. As a spice and food product, flavor is an important feature of *A. villosum*. Furthermore, we performed metabolomic and flavoromic analyses of the seed clusters of the two *A. villosum* varieties. We discovered that A12 exhibits a higher expression of terpenoid metabolites and flavor compounds associated with fruity, floral, and green, potentially explaining its richer flavor and better taste. This also indicates a significant correlation between the metabolic differences of *A. villosum* and its flavor. Moreover, using an electronic nose and electronic tongue, Liu et al. [29] found a significant correlation between the aroma of *A. villosum* and the quality of its medicinal ingredients, suggesting that flavor may serve as a convenient sensory characteristic reflecting the intrinsic quality of *A. villosum*, thus providing insights for variety breeding and rapid quality assessment.

Furthermore, we analyzed the volatile oil content and the main active ingredient content in the two varieties. We found no significant differences in the volatile oil content and borneol content between the two varieties, but significant differences were observed in the contents of borneol acetate and camphor. Borneol acetate and camphor are the two most important active terpenoid secondary metabolites in *A. villosum* [22]. Borneol is transformed into borneol acetate under the catalysis of borneol acetyltransferase and into camphor under the action of borneol dehydrogenase [30,31]. In the comparison of the two varieties, while the borneol content is almost the same in both, the borneol acetate content in A12 is significantly higher than that in A11, whereas the camphor content is significantly higher in A11 than in A12. Therefore, we speculate that there may be different regulatory mechanisms in the process of borneol transformation into these two substances in different varieties, and borneol acetyltransferase and borneol dehydrogenase may be in a competitive relationship. The vastly different characteristics of the two varieties also provide essential material foundations for the development of camphor-type *A. villosum* (high camphor content) and borneol acetate-type *A. villosum* (high borneol acetate content).

The same genome, but with differences in gene expression regulation, is the main reason for the trait differences among varieties [32]. We performed an integrated transcriptomic and metabolomic analysis of the two varieties and found significant differences in the synthesis and accumulation processes of secondary metabolites between them. In 2022, the chromosome-level reference genome sequence of *A. villosum* was publicly released [29]. This study combined genomics, transcriptomics, metabolomics, and in vitro enzyme function identification to reveal that most terpenoid synthases are multi-product enzymes that are highly expressed in seed clusters [29]. Further leveraging this genome, Liang et al. identified eight genes that catalyze the conversion of different types of borneol to borneol acetate [6]. Additionally, Chen et al. identified ten candidate borneol dehydrogenase (BDH) genes that may catalyze the conversion of borneol to camphor in their genomic and transcriptomic study of three original species (variants) of *A. villosum* [33]. Unfortunately, the functions of these genes have not been effectively validated within *A. villosum*. Furthermore, the transcription factors critical for the regulation of these gene expressions remain unidentified. Therefore, the molecular mechanisms governing the synthesis of borneol, bornyl acetate, and camphor in *A. villosum* require extensive scientific investigation. This study identified a gene associated with the synthesis and accumulation of bornyl acetate, which may account for the significant differences in bornyl acetate content between the two varieties. However, more functional validation studies are necessary to draw more definitive conclusions.

## 4. Materials and Methods

### 4.1. Plant Material

A11 is a widely cultivated farm variety, commonly referred to as “Huangmiaozi.” A12, initially a mutant specimen, was collected in 2012 from an *A. villosum* plantation in Panlong Village, Chuncheng District, Yangchun City, Guangdong Province. Subsequently, it was transferred to a resource conservation garden for *A. villosum* in Mazhang District, Zhanjiang City, Guangdong Province, where it was preserved and propagated vegetatively, following the guidelines outlined in the “*A. villosum* Cultivation Technical Specification” (T/GDSMM 0013-2021) [34]. Starting in 2014, asexual reproduction was employed to evaluate the plant morphology, growth, and yield characteristics of A12, with generation-by-generation tracking. The results indicate stable genetic inheritance across generations, with consistent botanical characteristics, yield, and quality from year to year, demonstrating good stability and consistency. In this study, A11 and A12 were introduced to the experimental site in March 2021 (Jinhua Keng, Panlong Town, Yangchun City, Guangdong Province, China; 22.19° N, 111.93° E). The experiment was conducted using a randomized complete block design, with each variety planted in 3 plots. Each plot measured 60 m^2^ (6 m × 10 m), with 30 *A. villosum* plants planted in each plot, and subjected to identical field management practices. Plant characteristics were observed in September 2023, and fruit and other experimental samples were collected for subsequent analysis.

### 4.2. DNA Barcode Molecular Identification

An amount of 0.5g of fresh leaf tissue from two random varieties of *A. villosum* was taken, ground thoroughly with liquid nitrogen, and centrifuged, and total DNA was extracted using the SDS method. The ITS2 universal primers (forward primer: ITS5F: 5′-GGAAGTAAAAGTCGTAACAAGG-3′, reverse primer: ITS4R: 5′-TCCTCCGCTTATT GATATGC-3′) specified in the “DNA Barcode Standard Sequence of Chinese Pharmacopoeia” were used to amplify specific fragments. The obtained PCR products were sent to Sangon Biotech (Shanghai) Co., Ltd. (Shanghai, China) for bidirectional DNA sequencing. The obtained sequences were then imported into the ITS2 database website (http://its2.bioapps.biozentrum.uni-wuerzburg.de/ accessed on 18 February 2024) for sequence annotation to obtain complete ITS2 sequences. The ITS2 sequences of *A. villosum*, related plants, and adulterants were retrieved from NCBI: *A. villosum* (GenBank, AF478724.1), *Amomum longiligulare* (LC583276.1), *Amomum muricarpum* (OK161309.1), *Amomum gagnepainii* (LC583280.1), *Elettaria cardamomum* (OR786978.1), and *Alpinia oblongifolia* (OR786958.1). The obtained ITS2 sequences were aligned with the *A. villosum* standard sequences (CK), genetic distances were calculated, and a neighbor-joining (NJ) phylogenetic tree was constructed.

### 4.3. Investigation of Plant Morphology and Medicinal Characteristics

Referring to the records in “Flora of China”, an investigation was conducted into the morphological characteristics of the roots, rhizomes, flowers, leaves, fruits, and seeds of the two varieties. Following the descriptions of the characteristics of Amomi Fructus in the “Chinese Pharmacopoeia” (2020 edition), an investigation was conducted into the characteristics of Amomi Fructus produced by the two varieties. In the survey, 10 plants were randomly selected from each plot for observation, and the average values were calculated. The yield is converted to yield per acre based on the actual yield of each plot.

### 4.4. Determination of Active Ingredient Content

The volatile oil content was determined according to the method specified in the “Chinese Pharmacopoeia” 2020 edition (General Rule 2204). Using a GC-MS (gas chromatography–mass spectrometry) instrument (Agilent 7890B-5977B; Agilent, Beijing, China), with bornyl acetate, camphor, and borneol as standard reference solutions, the content of the three main active ingredients in the two varieties of *A. villosum* was determined. The main parameters of gas chromatography were as follows: high-purity helium gas used as the carrier gas; temperature program set as follows: initial temperature at 40 °C, held for 10 min, then ramped at 10 °C/min to 150 °C, held for 5 min, further ramped at 20 °C/min to 280 °C, held for 5 min; injector temperature set at 230 °C; injection volume of 2 μL; split injection with a split ratio of 10:1; and transfer line temperature of 250 °C. The main parameters of the mass spectrometry were EI ionization source; ionization energy of 70 eV; and source temperature of 200 °C.

### 4.5. Metabolomics Testing

#### 4.5.1. Detection of Metabolites Based on the LC-MS Platform

The samples were placed in a freeze dryer (Scientz-100F; Scientz, Ningbo, China) for lyophilization. After the samples were completely dried, they were ground into a powder using a grinder (MM 400, Retsch, Shanghai, China) at 30 Hz for 1.5 minutes. Next, 50 mg of sample powder was weighed using an electronic balance (MS105DΜ, Mettler Toledo, Shanghai, China), and 1200 μL of −20 °C pre-cooled 70% methanolic aqueous internal standard extract was added. This was vortexed once every 30 min for 30 s, for a total of 6 times. After centrifugation (rotation speed 12,000 rpm, 3 min; 22331, Eppendorf AG, Hamburg, Germany), the supernatant was collected, and the sample was filtered through a microporous membrane (0.22 μm pore size, Membrane Solutions, Texas, USA) and stored in the injection vial for UPLC-MS/MS analysis.

The sample extracts were analyzed using a UPLC-ESI-MS/MS system (UPLC, ExionLC AD, SCIEX, Shanghai, China; MS, Applied Biosystems 6500 Q TRAP, Applied Biosystems, Shanghai, China). The effluent was alternatively connected to an ESI-triple quadrupole-linear ion trap (QTRAP)-MS. QQQ scans were acquired as MRM experiments with collision gas (nitrogen) set to medium. DP (declustering potential) and CE (collision energy) for individual MRM transitions were conducted with further DP and CE optimization. A specific set of MRM transitions was monitored for each period according to the metabolites eluted within this period. Qualitative and quantitative analyses of the substances were performed based on the MWDB (Metware database) and secondary spectrum information.

#### 4.5.2. Detection of Metabolites Based on the GC-MS Platform

The samples were ground to a powder in liquid nitrogen. Then, 0.2g of the powder was transferred immediately to a 20 mL headspace vial (Agilent, Palo Alto, CA, USA) containing 0.2 g NaCl powder, to inhibit any enzyme reaction. After sampling, desorption of the VOCs from the fiber coating was carried out in the injection port of the GC apparatus (Agilent 8890, Agilent, Beijing, China) at 250 °C for 5 min in the splitless mode. The identification and quantification of the VOCs were carried out using an Agilent Model 8890 GC and a 7000E mass spectrometer (Agilent, Beijing, China), equipped with a 30 m × 0.25 mm × 0.25 μm DB-5MS (5% phenyl-polymethylsiloxane) capillary column. The mass spectrometry was conducted using selected ion monitoring (SIM) mode. Further qualitative and quantitative analyses were performed based on multi-species, the literature, partially labeled compounds, and a self-built database with retention indices.

### 4.6. Flavoromics Analysis

Based on the GC-MS detection, the widely-targeted volatolomics 2.0 (WTV 2.0) method developed by Wuhan Metware Metabolic Biotechnology Co., Ltd. (Wuhan, China) was employed, utilizing the retention index (RI) and odor database, Flavornet, human odor space, and Flavor Ingredient Library to annotate the odor of the substances [35]. Furthermore, the relative odor activity value (rOAV) analysis was conducted on the substances with flavor characteristics, and the analysis results were further presented in flavor radar charts, mulberry charts, and other visualizations.

### 4.7. Transcriptomic Analysis

Total RNA was extracted from the samples using ethanol precipitation and CTAB–PBIOZOL methods, and the quality and quantity of total RNA were identified using the Qubit fluorescence quantitation instrument and the Qsep400 high-throughput biological fragment analysis instrument. The mRNA with a polyA tail was enriched using Oligo(dT) magnetic beads and further constructed into a cDNA library. The library quality was detected and the sequencing was performed on the Illumina platform after the results met the requirements (effective concentration > 2 nM). The raw data obtained after sequencing was filtered to obtain clean data, which was aligned with the *A. villosum* reference genome [22] to obtain mapped data. Structural-level analyses, such as alternative splicing analysis, novel gene discovery, and gene structure optimization, were performed on the mapped data. Expression-level analyses, such as differential expression analysis, functional annotation of differentially expressed genes, and functional enrichment, were performed according to the expression levels of genes in different groups.

### 4.8. Data Analysis

Data organization, significance analysis, radar chart analysis, etc., were conducted using software such as Excel (Office 16), SPSS (22), and Origin (2020). R software (3.6.1) and the Metware Cloud Platform (https://cloud.metware.cn/ accessed on 20 December 2023) were utilized for multivariate statistical analysis and visualization processing of the metabolomics and transcriptomics data, including principal component analysis, hierarchical clustering analysis, correlation analysis, and other analyses.

## 5. Conclusions

Compared to A11, A12 exhibits several superior traits, including more robust plant growth, a more developed root system, increased rhizome branching, larger and more numerous inflorescences, larger seeds, stronger aroma, better taste, higher yield, and higher borneol acetate content. These characteristics demonstrate A12’s potential as an elite variety. Conversely, A11 shows potential for developing into a variety with high camphor content. Further multi-omics analyses revealed that A12 has more upregulated terpenoid metabolites and flavor compounds associated with fruity, floral, and green notes, which may explain its stronger aroma and better taste. Additionally, the biosynthesis of various plant secondary metabolites was identified as the main pathway enriched in the differential metabolites between the two varieties, indicating significant differences in their secondary metabolite biosynthesis pathways. Correlation analysis and qPCR validation confirmed that the key gene regulating borneol acetyltransferase (Wv_032842) is significantly upregulated in A12, which may account for the higher borneol acetate content in this variety. In summary, this study assessed the germplasm of the two *A. villosum* varieties from multiple perspectives and used integrated multi-omics analysis to elucidate the reasons behind their differences. The findings offer valuable insights into the breeding and quality improvement of *A. villosum* varieties.

## Figures and Tables

**Figure 1 plants-13-02382-f001:**
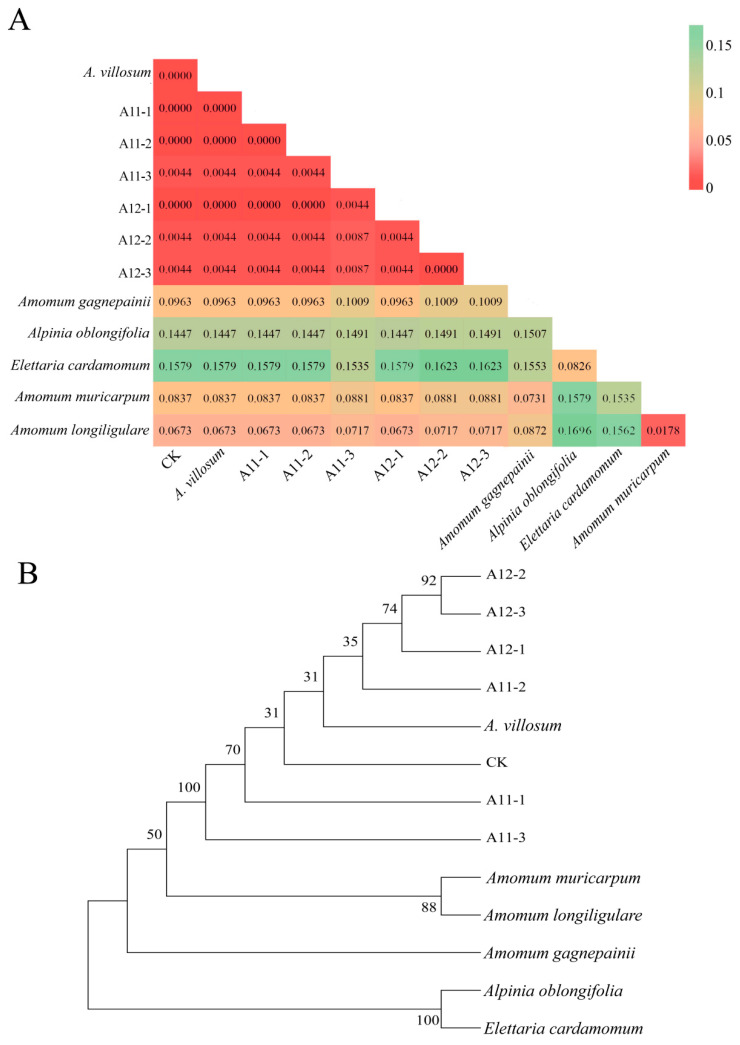
DNA barcode analysis of A11, A12, and closely related plants: (**A**), genetic distance analysis of A12 and A11 relative to closely related plants based on ITS2 sequence; (**B**), neighbor-joining (NJ) phylogenetic tree. CK is the standard sequence published by the “DNA Barcode Standard Sequence of the Chinese Pharmacopoeia”. A11-1, A11-2, and A11-3 are replicates of A11, while A12-1, A12-2, and A12-3 are replicates of A12.

**Figure 2 plants-13-02382-f002:**
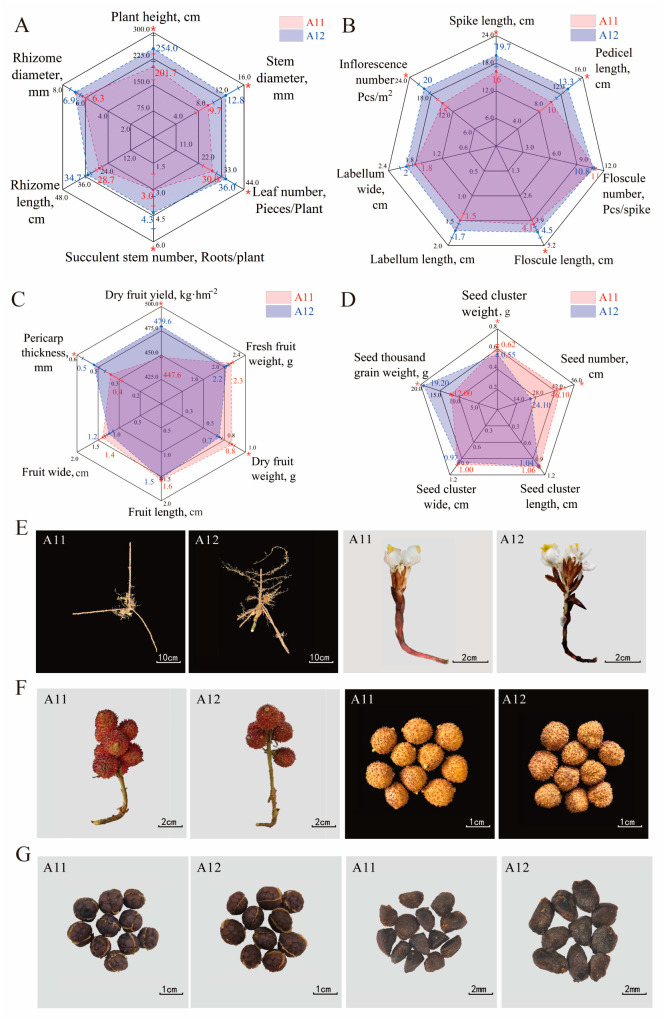
Comparison of phenotypic traits between A11 and A12: (**A**–**D**), illustration of the phenotypic measurements of *A. villosum* plants, flowers, fruits, and seed clusters (mean value of 3 replicate experiments); The “*” denotes statistically significant differences between groups at the *p* < 0.05 level. (**E**), the rhizome and inflorescence diagram; (**F**), the images of fresh and dried fruits; (**G**), the seed clusters and seeds.

**Figure 3 plants-13-02382-f003:**
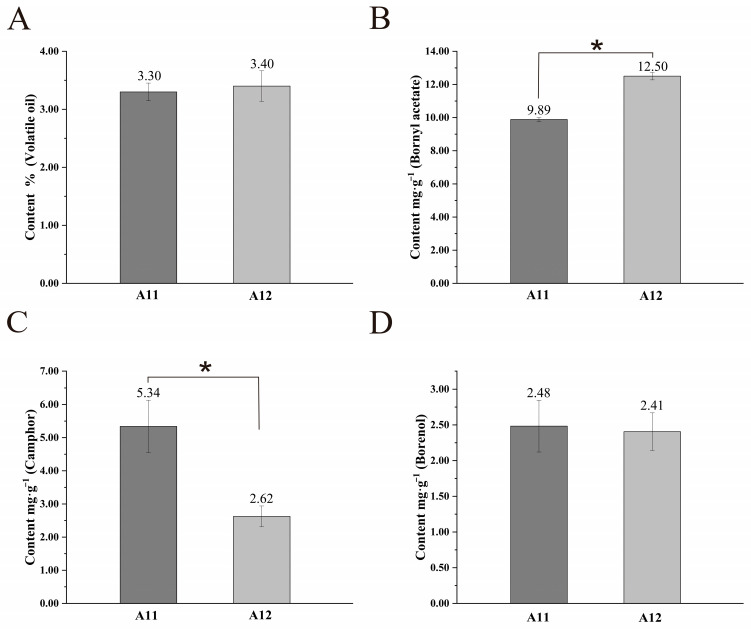
Comparison of the content of major active ingredients between A11 and A12 (*n* = 3): (**A**), total volatile oil; (**B**), bornyl acetate; (**C**), camphor; (**D**), borneol. The “*” denotes statistically significant differences between groups at the *p* < 0.05 level. “n” represents the number of biological replicates.

**Figure 4 plants-13-02382-f004:**
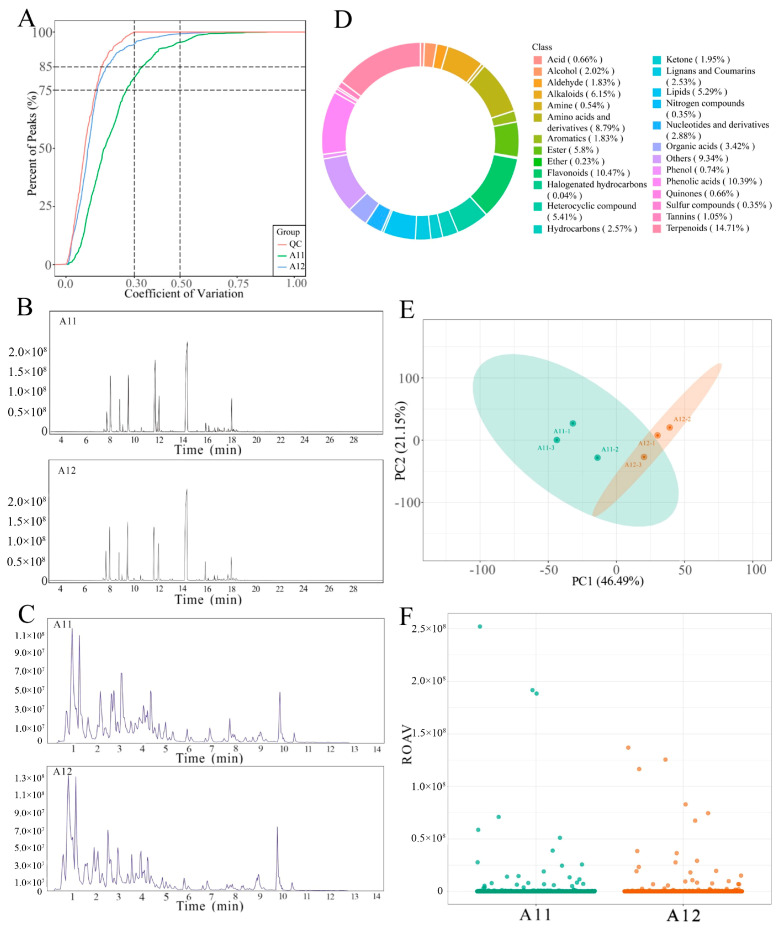
The metabolomic analysis of A11 and A12: (**A**), distribution of CV; (**B**), TIC chromatogram of GC-MS; (**C**), TIC chromatogram of LC-MS/MS (pos); (**D**), circular chart of substance classification; (**E**), metabolomic PCA plot; (**F**), scatter plot of rOAV odor activity values.

**Figure 5 plants-13-02382-f005:**
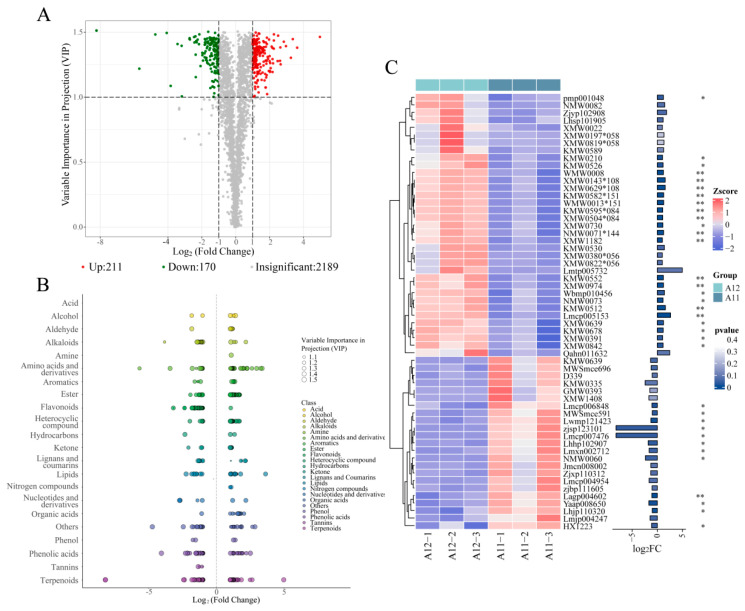
Differential metabolite analysis of A12 vs. A11: (**A**), volcano plot of differential metabolites; (**B**), scatter plot of differential metabolites, with each point representing a metabolite; (**C**), terpenoid differential metabolite cluster heatmap and bar plot. The “*” denotes statistically significant differences between groups at the *p* < 0.05 level; the “**” denotes statistically significant differences between groups at the *p* < 0.01 level.

**Figure 6 plants-13-02382-f006:**
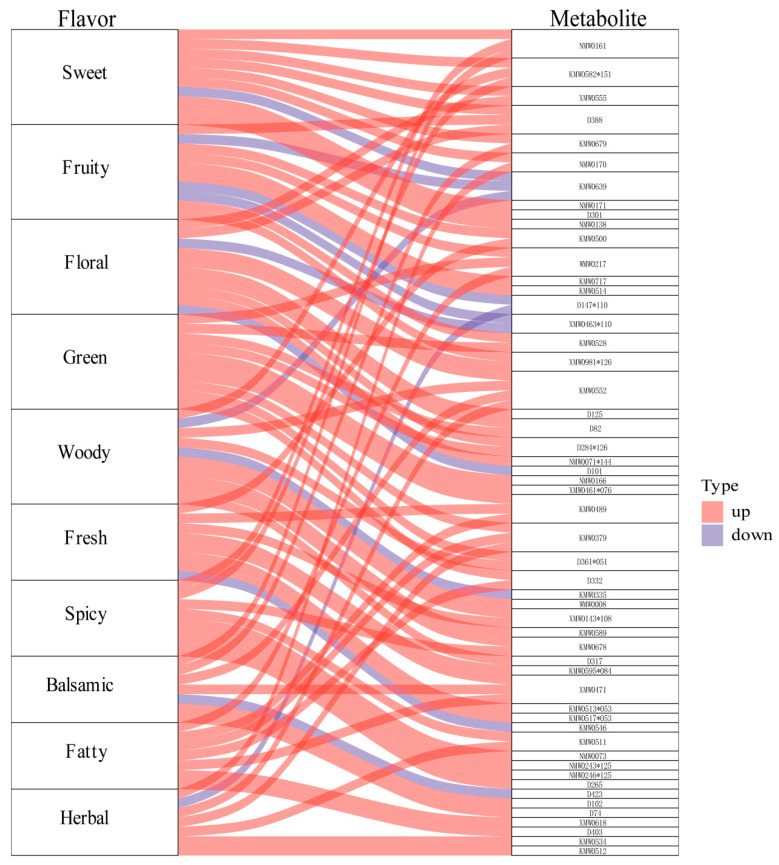
Sankey diagram analysis of differential flavor metabolites (TOP10): the red lines represent upregulated metabolites in A12 versus A11, while the blue lines represent downregulated metabolites in A12 versus A11.

**Figure 7 plants-13-02382-f007:**
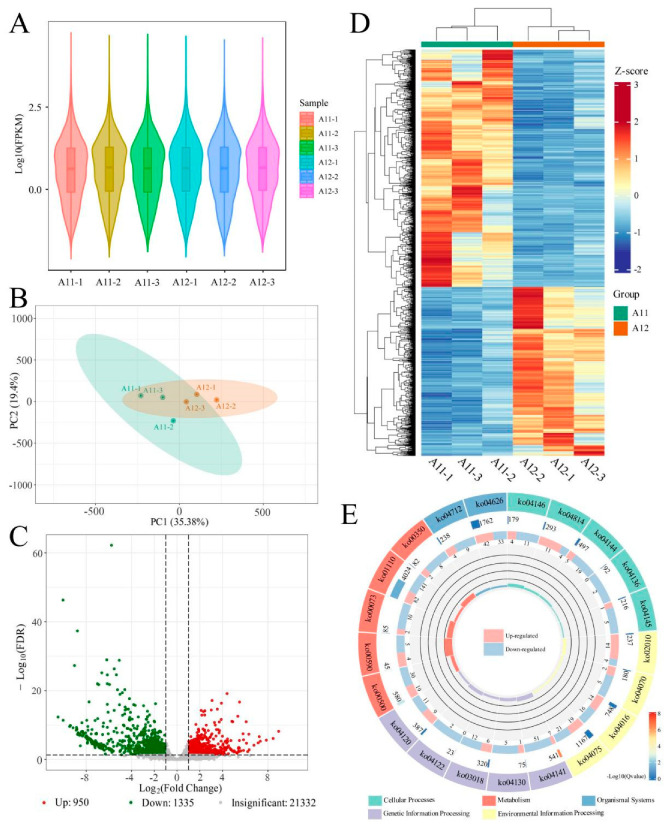
Transcriptomics and differential expression gene analysis between A11 and A12: (**A**), violin plots of gene expression levels; (**B**), PCA analysis of the transcriptome; (**C**), volcano plot of differentially expressed genes; (**D**), heatmap of clustered differentially expressed genes. (**E**) KEGG circle plot of differentially expressed genes. From outer to inner layers: the first layer represents KEGG_level_1 entries; the second layer displays the number of genes in each classification within the background genes and their q-values; the third layer shows a bar plot of the ratio of upregulated to downregulated genes; the fourth layer illustrates the RichFactor values for each classification, with background auxiliary lines indicating 0.2 for each small segment.

**Figure 8 plants-13-02382-f008:**
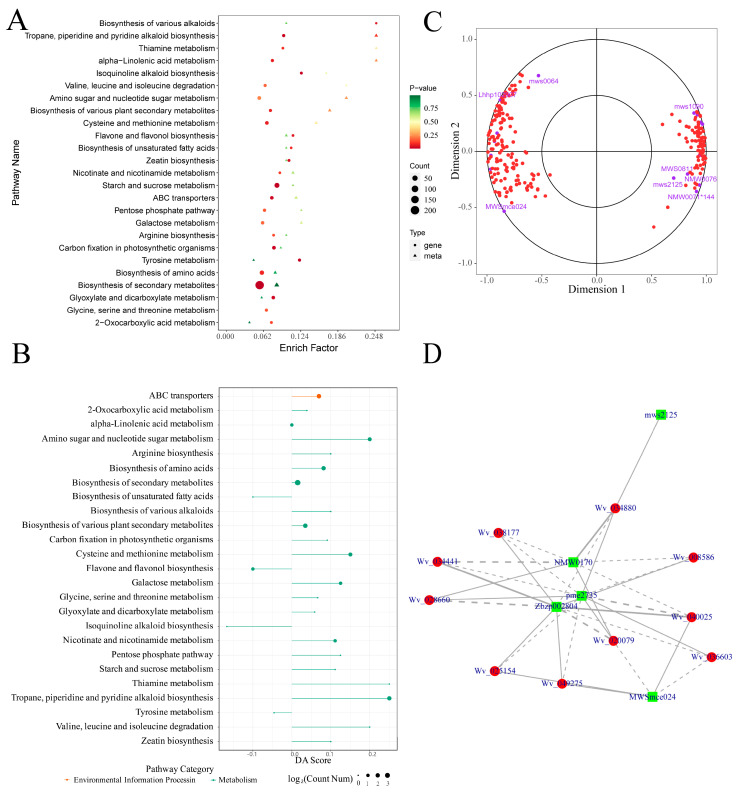
Transcriptome and metabolome correlation analysis: (**A**), bubble plot of KEGG enrichment analysis associating differentially expressed genes and metabolites; (**B**), differential abundance score plot (DA Score); (**C**), canonical correlation analysis (CCA) plot of differentially expressed genes and metabolites in the ko01110 pathway, where red circles represent genes and purple circles represent metabolites; (**D**), network diagram of the correlation between differentially expressed genes and metabolites in the ko00999 pathway, where red circles represent genes, green squares represent metabolites, solid lines indicate a positive correlation and dashed lines indicate a negative correlation.

**Figure 9 plants-13-02382-f009:**
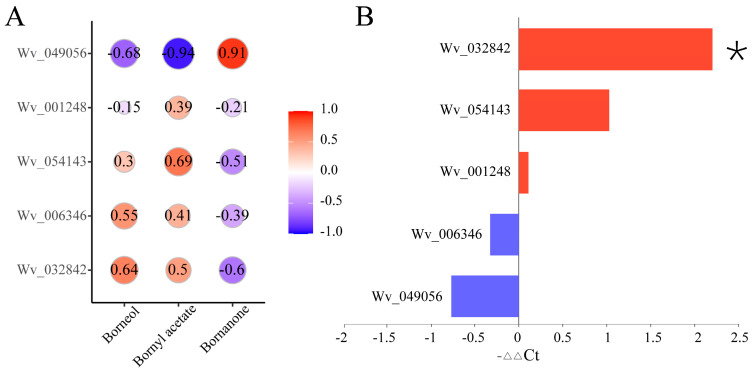
Correlation analysis of metabolites and key genes, and qPCR validation: (**A**), heatmap showing the correlation between key genes and borneol acetate, camphor, and borneol; (**B**), qPCR validation of key genes. The “*” denotes statistically significant differences between groups at the *p* < 0.05 level.

## Data Availability

Data are contained within the article and Appendix A.

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
