# Peer review of "Integrated Analysis of Metabolomics, Flavoromics, and Transcriptomics for Evaluating New Varieties of Amomum villosum Lour."

_plants, 2024, doi:10.3390/plants13172382_

Round 1
Reviewer 1 Report
Comments and Suggestions for Authors
In this study, the authors compare two varieties of Amomum villosum plants. The work merits publication, and it is crucial that the authors have clarified the following parts of the manuscript previously, as this will significantly enhance the value of their research.
Describe how the long-term selection of the two varieties A-11 and A-12 was made regarding the parent wild-type.
Related to the previous question, the authors mention that the wild-type plant has a low yield. However, they must provide data for comparison, as this will strengthen their argument and provide a more comprehensive analysis. The authors give data of 10.0 kg/667 m2 of dried fruit of the parent plants. However, they report data on their new varieties regarding fresh weight, so the comparison is difficult.
In their methodology, the authors mention that yield will be calculated in acres instead of international metrics, such as hectares. Furthermore, the authors give results in m2, and more specifically, by 667 m2; from where did this calculation come?
Some discussion paragraphs seem more suitable for the introduction section: 299-310 and 370-378.
One crucial point: the authors should have included some parental control in their omics analysis. This control is critical to show the value of their variety selection.
The authors mention the A12 as a potential elite variety, but they should explain how to manage genetic markers for this aim. The authors did not analyze the biosynthetic pathways for camphor or borneol synthesis to show where the differences between the plants are.
Reviewer 2 Report
Comments and Suggestions for Authors
The purpose of the study is clear: to evaluate new varieties of A. villosum, to ensure diversity and to contribute to further breeding in the future. While the experiment is clear, I found some difficulty in the way it was presented. The quality of the figures showing the results is low and insufficient to present the results to the reader. Many of the expressions in the text are not meaningful, as they merely list the indices obtained in the course of the analysis. It would be easier for readers to understand if more general expressions such as substance names and gene names were used first, instead of using headings in the analysis application. I got the impression that the quality of the article was deteriorating due to inadequate expression methods, even though useful information seems to have been obtained. Significant improvement of the expression is requested.
Major comments; section 2.2, are those results the only one trial or those values are mean of several trials? If those results are means of measurements, please indicate errors, or show those results obtained from how many measurements.
In Fig3B, y-axis label is “content mg g-1 (Bornyl acetate), and A11 9.89, A12 12.50. However in text line 144–147, the content of ethyl lauroyl alginate in A12 (12.5 mg g-1). Are Bornyl acetate and ethyl lauroyl alginate the same compound or different compound? (arginate means alginate or arginine?)
Please check text or axis label. Additionally, star means what? Describe in detail.
This is my opinion, the author uses the analytical index name in preference to the substance name, but it is preferable to use the compound/substance name in preference to the analytical heading. (Please check overall the manuscript)
Regarding this suggestion, line 211 “KMW0582*151 (.beta.-Guaiene) “ , remove periods then describe β-Guaiene.
Section 3.5; It is unclear what section 3.5 is referring to. It says that the top 10 sensory flavor annotations were selected, but is it talking about the top 10 for conventional A.villosum (not A11 or A12), or is it talking about A11 or A12, or is it talking about the components that were detected more in A12 than A11? The subject matter is ambiguous and difficult to understand; Figure 6B is also uninformative due to the small font size and insufficient resolution.
The author is working on a transcriptome and has mentioned borneol acetyltransferase and borneol dehydrogenase in the discussion. The changes in transcripts of these enzymes should be shown in the results and discussed in the discussion while showing the results.
Minor comments; Figure 1, please explain the listed labels (YCSR, CXSR, HSJ, XDK, YGSR, and HNSR) in figure legend. It would be better to use the botanical name rather than listing it as YCSR or HNSR. Furthermore, I felt that overall there was not enough information about figure 1 (e.g., what are the values of figure 1A and 1B). Are these three A11-1, A11-2, A11-3 triplicates or different sub species?
Sharper images (Figures 2A-D), text size, and resolution quality should be improved. This is my opinion, but how about grouping Figures 2C and 2D together as fruits and seeds? Figures 2C and 2D have 6 and 5 labels, respectively: 2C has 3 labels for fruit and 3 labels for seeds; 2D has 3 labels for fruit (including pericarp) and 2 labels for seeds. The 6 labels for fruits are then summarized as FIg2C and the 5 labels for seeds as Fig2D.
Section 5.2; A. Villosum is italic but the other not, then it is better to write italic.
Section 5.51; line 459–462, About centrifugation, describe as centrifugal force (× g) or what equipment was used (rotor number and equipment name, or radius with rotation speed)because the centrifuged condition is unclear. “The supernatant was aspirated,” means the supernatant was collected? When described as “the supernatant was aspirated,” I feel that author wants to obtain the remaining pellets.
Figure 4 and 5; It is better that plot axis and text etc, more clearly (dark color, font size up, etc). Some plots are difficult to understand the informations.
Line 341, Probably, missing period after “utilization”, or “As” => “as”.
Round 2
Reviewer 1 Report
Comments and Suggestions for Authors
The authors responds to all my observations, now the manuscript is much better.
I could observe a mistake in line 153, it lacks kg in ... at 447.61 kg.hm
Reviewer 2 Report
Comments and Suggestions for Authors
We felt that the areas we pointed out had been appropriately revised and were of a quality appropriate for publication. We look forward to further research breakthroughs in the future.
